# Clinical Characteristics of the COVID-19 Patients with Pneumonia Detected by Computerized Tomography but Negative for Infiltration by X-ray

**DOI:** 10.3390/healthcare8040518

**Published:** 2020-11-29

**Authors:** Dilaram Acharya, Jungi Park, Yebong Lee, In Suk Hamm, Dong Seok Lee, Seong-Su Moon, Kwan Lee

**Affiliations:** 1Department of Preventive Medicine, College of Medicine, Dongguk University, Gyeongju 38066, Korea; dilaramacharya123@gmail.com; 2Department of Community Medicine, Devdaha Medical College and Research Institute, Kathmandu University, Rupandehi 32900, Nepal; 3Department of Internal Medicine, College of Medicine, Dongguk University, Gyeongju 38066, Korea; pogonion@naver.com; 4Department of Internal Medicine, Pohang Medical Center, Pohang 37688, Korea; Yebong-lee@hanmail.net; 5Department of Neurosurgery, Pohang Medical Center, Pohang 37688, Korea; ishamm@knu.ac.kr; 6Department of Pediatrics, College of Medicine, Dongguk University, Gyeongju 38066, Korea; lds117@dongguk.ac.kr; 7Section of Endocrinology, Department of Internal Medicine, Nazareth General Hospital, Daegu 42784, Korea

**Keywords:** COVID-19, computerized tomography, pneumonia, risk factors, SARS-CoV-2

## Abstract

Coronavirus Disease 2019 (COVID-19) has rapidly spread to all corners of the globe. Different diagnostic tools, such as Chest X-ray (CXR), lung ultrasound (LUS), and computerized tomography (CT), have been used to detect active pneumonic lesions associated with COVID-19 with their varying degrees of sensitivity and specificity. This study was undertaken to investigate the clinical characteristics of COVID-19 patients with a pneumonic lung lesion detected by CT that is not detected by CXR. A total of 156 COVID-19 patients hospitalized at three nationally designated South Korean hospitals with no active lesion detected by CXR but on clinical suspicion of pneumonia underwent the CT examination and were enrolled. Medical records, which included demographic and clinical features, including comorbidity, symptoms, radiological, and laboratory findings on admission, were reviewed and analyzed. The risk factors of pneumonia detected by CT for patients without an active lesion detected by CXR were investigated. Of the 156 patients without an active lesion detected by CXR, 35 (22.44%) patients were found to have pneumonia by CT. The patients with pneumonia defined by CT were older than those without (64.1 years vs. 41.2 years). Comorbidities such as hypertension, diabetes, cardiovascular disease, preexisting stroke, and dementia were more common among patients with pneumonia defined by CT than those without. Serum albumin level, C-reactive protein (CRP), stroke, and age ≥ 70 years were significantly associated with pneumonia defined by CT after adjustment for age. In multivariable regression analysis, serum albumin level (adjusted odds ratio (AOR) = 0.123, 95% CI = (0.035–0.429)) and preexisting stroke (AOR = 11.447, 95% CI = (1.168–112.220)) significantly and independently predicted pneumonia detection by CT. Our results suggest that CT scans should be performed on COVID-19 patients negative for a pneumonic lung lesion by CXR who are suspected to be pneumonic on clinical grounds. In addition, older patients with a lower albumin level and a preexisting stroke should be checked for the presence of pneumonia despite a negative CXR finding for an active lesion.

## 1. Introduction

The first cases of Corona Virus Disease 2019 (COVID-19) were detected in Wuhan, Hubei Province, China in December 2019, and COVID-19 rapidly reached pandemic status; 35,701,674 cases and 1,045,953 deaths worldwide were attributed to the disease on 10 October 2020 [1,2]. In South Korea, according to the Ministry of Health and Welfare, 24,703 confirmed cases and 433 deaths were reported as of 12 October 2020 [3].

The disease presents with characteristics that range from asymptomatic states to lethal conditions such as acute respiratory distress syndrome (ARDS), severe pneumonia, acute kidney injury, myocarditis, and even multi-organ failure or death [4,5,6,7]. In addition to intolerable physical and mental health problems, the disease has resulted in severe socio-economic disruption and reduced quality of life worldwide [8,9,10,11]. Unfortunately, unless an effective vaccine or therapy is developed, the pandemic will, in all likelihood, continue [12].

Several studies have reported poor health outcomes for COVID-19 patients with accompanying conditions such as respiratory, cardiac, or renal diseases or diabetes [13,14,15], and the obese and elderly are commonly reported to be at high risk of serious disease [15,16]. Approximately, 15–20% of all those infected (cases) develop severe pneumonia, 5–10% need critical care, and pneumonic COVID-19 patients are more likely to succumb to the disease than those with pneumonia of another etiology [14,17]. Furthermore, concerns have been expressed regarding increasing numbers of false positive real-time polymerase chain reaction (RT-PCR) and sequencing test results, which means missed infections, greater transmission opportunities, and poorer prognoses. Therefore, it is worth considering whether the diagnosis of pneumonic COVID-19 should be based on chest X-ray (CXR), CT, or RT-PCR findings [18]. Although CXR is commonly used to detect active pneumonia lesions, CT has advantages over CXR for the diagnosis of COVID-19 associated with chest symptoms and has higher sensitivity and specificity than CXR in patients with COVID-19 [19].

In this study, we examined the clinical and laboratory data of COVID-19 patients treated at three South Korean tertiary level hospitals with a pneumonic lung lesion detected by CT but not detected by CXR to identify the clinicopathological features of this patient population and the clinical characteristics and associated risks of hidden COVID-19 pneumonia.

## 2. Materials and Methods

### 2.1. Study Subjects

Among 679 patients with a confirmed diagnosis of COVID-19 infection hospitalized at three nationally designated hospitals for COVID-19, namely, Dongguk University Gyeongju Hospital (DUGH), Pohang Medical Center (PMC), and Andong Medical Center (AMC), Gyeongsangbuk-do, South Korea between 18 February and 30 June 2020, 156 patients who had no active lesion on chest X-ray (CXR) but who were on clinical suspicion of pneumonia underwent computerized tomography (CT) for further evaluation on admission and were enrolled in the present study. Chest radiology images were examined by radiologists at the three hospitals. For patients transferred to DUGH from PMC or AMC, initial PMC or AMC records were included to prevent duplication. Information on demographic and clinical characteristics, including CXR, CT, and laboratory findings were collected. Details of pre-existing diseases, vital signs (i.e., blood pressure, heart rate, respiratory rate, and body temperature), and pulse oximetry oxygen saturation in room air were obtained at admission. Laboratory findings for blood sampled at admission included complete blood cell (CBC) count, aspartate aminotransferase (AST), alanine aminotransferase (ALT), blood urea nitrogen (BUN), creatinine, serum albumin, C-reactive protein (CRP) level, and erythrocyte sedimentation rate (ESR).

### 2.2. Statistical Analysis

Nominal variables are presented as numbers of cases and percentages, and continuous variables as means and standard deviations (SDs) or ranges. Patients were categorized into two groups: “no active lesion on CXR and CT” or “no active lesion on CXR but active infiltration on CT”. Student’s *t*-test or the Mann-Whitney U test was used to determine the significance of the differences between these two groups. χ2 analysis or Fisher’s exact test was used to determine the significance of differences between categorical variables. Univariate and multivariate logistic regression analyses were performed to investigate associations between clinical and laboratory variables and the risk of a pneumonic lung lesion as determined by CT (PLLCT). The potential risk factors included in the univariate analysis were age ≥70 years, care facility resident, presence of a comorbidity (i.e., hypertension, diabetes, dementia, or stroke), physical examination findings (i.e., systolic blood pressure ≤90 mmHg and pulse oximetry oxygen saturation in room air ≤90 mmHg), and laboratory findings (i.e., AST, hemoglobin, albumin, total protein, ESR, and CRP). Multivariate logistic regression analysis was used to determine the nature of associations between potential risk factors identified by univariate analysis. All tests were two-sided, and statistical significance was accepted for *p* values < 0.05. The analysis was performed using the Statistical Package for Social Sciences version 20.0 (SPSS, Chicago, IL, USA).

### 2.3. Ethics Statement

This study protocol was exempted from ethical review by the Institutional Review Board of Dongguk University Gyeongju Hospital (IRB registration No. 110,757–202,006-HR-04-02) because the data analyzed did not contain information that might be used to identify individuals.

## 3. Results

### 3.1. Characteristics of the Hospitalized Patients

The clinical and laboratory profiles of the COVID-19 study subjects are presented respectively, in Table 1 and Table 2. Of the 156 study subjects, 22.44% (35/156) had pneumonic lung lesions defined by computerized tomography (PLLCT) not visualized by CXR. The mean age of those pneumonic subjects with lung lesions (the PLLCT group) was older than that of those without (64.1 ± 15.5 vs. 41.2 ± 18.4), and days of hospitalization and death rates were significantly higher in the PLLCT group (*p* < 0.05). Furthermore, the PLLCT group had a higher prevalence of comorbidities such as hypertension, diabetes mellitus, dementia, cardiovascular disease, and stroke (*p* < 0.05). Laboratory testing showed that aspartate aminotransferase, creatinine, the erythrocyte sedimentation rate, and C-reactive protein were significantly higher but that hemoglobin, albumin, and total protein were significantly lower in the PLLCT group (*p* < 0.05).

### 3.2. Risk Factors for Pneumonia in COVID-19 Patients with a Pneumonic Lung Lesion Detected by CT

The univariate regression analysis showed that an age ≥70 years, a history of stroke, elevated C-reactive protein, and a low albumin level were significantly related to the presence of PLLCT (Table 3). Multivariate regression analysis revealed that a history of stroke (adjusted odds ratio (AOR) = 11.447, 95% CI = (1.168–112.220)) was significantly related to PLLCT and that a reduction in albumin by 1 mg/dL was associated with a 0.123 increase in the risk of PLLCT (AOR = 0.123, 95% CI = (0.035–0.429)), as in Table 4.

## 4. Discussion

To the best of our knowledge, this study is the first to describe the characteristics of hospitalized Korean COVID-19 patients without an active pneumonic lesion detected by CXR but with an active lesion detected by CT. Thirty-five of the 156 study subjects (22.44%) were found to have PLLCT. In addition, adjusted multivariate logistic regression analysis showed that only two variables, namely, serum albumin level and stroke history, significantly predicted the presence of PLLCT.

In the current pandemic situation of COVID-19, the use of effective and efficient diagnostics is of paramount importance in the prevention and control of the disease. The more commonly used diagnostic tools in the detection of COVID-19-associated pneumonic lungs are CXR, lung ultrasound (LUS) and CT scans, with their varying degree of sensitivity and specificity [20]. In addition, there are obviously pros and cons of these different diagnostic tools in terms of detectability, feasibility, and cost considerations [20,21]. For instance, compared with CT scans and CXR, LUS lacks exposure to radiation and, thus, might be particularly useful in critical care settings and for pregnant women, children, and patients in areas with high rates of community transmission of SARS-CoV-2 infection [21]. However, previous studies have reported that CT is instrumental in terms of reducing false negatives among pneumonic COVID-19 cases when implemented two or more days after symptom development [18,22]. Furthermore, it has been reported that the sensitivity of CT for the diagnosis of COVID-19 pneumonia is ~97%, that that of initial RT-PCR is ~83% [22], and that CXR may be negative or non-diagnostic for pneumonia, whereas CT may depict definitive infiltrate consistent with pneumonia [23]. Given that COVID-19 usually has a short incubation period (between 2 to 10 days but no longer than 14 days) and is highly infective [24], these findings support the use of multiple diagnostics to reduce missed case rates and, thus, reduce transmissibility and adverse prognosis and mortality rates.

In the present study, PLLCT was predicted by hypoalbuminemia, which has been reported to be a significant predictor of malnutrition [25], pneumonia [26], all-cause mortality, and long-term hospital stays in severely ill patients [27]. Associations between hypoalbuminemia and increased disease severity and mortality among COVID-19 patients were also consistently observed in many earlier studies; for example, in a systematic review and meta-analysis [28], in a cross-sectional study [29], and in a cohort [30] study. In addition, a case series study conducted in Italy also concluded that a low albumin level was a significant risk factor for mortality and protracted hospital stay among hospitalized COVID-19 patients [31]. This association might be explained by a COVID-19-induced cytokine storm leading to hepatotoxicity, critical hypoalbuminemia, the exacerbate of disease-associated inflammatory responses, and disease progression [29]. Thus, it appears that inflammatory responses might be responsible for low serum albumin levels in COVID-19 patients with pneumonia [32]. However, we were unable to find a study that presented biologically plausible reasons for why hypoalbuminemia predicts the presence of PLLCT in COVID-19 patients. Nonetheless, current evidence shows the importance of investigating the possible presence of a pneumonic lesion using multiple technologies at the incipient disease stage to reduce morbidity and mortality and prevent COVID-19 transmission, especially in healthcare settings. Researchers have recently confirmed that the SARS-CoV-2 virus can circulate in blood, interact with ACE2 receptors expressed in brain capillary endothelium, and potentially damage the blood-brain barrier [33]. COVID-19 patients with pre-existing cerebrovascular disease have poor prognoses. A meta-analysis showed that these COVID-19 patients have a 2.67-fold (1.75 to 4.06) higher risk of a poor outcome [34]. Similarly, we also found that a preexisting stroke significantly predicted the presence of PLLCT. Therefore, we recommend that procedures should be devised and instituted to effectively manage COVID-19 patients that present with a cerebrovascular accident [34,35].

Our univariate analysis identified that the presence of comorbidities like hypertension, diabetes, cardiovascular disease, preexisting stroke, or dementia was significantly associated with PLLCT, compared to patients without. In line with these findings, Li et. al. [36] also reported that comorbidities were risk factors for severe/critical COVID-19 pneumonia as determined by CT, but, in the present study, the significance of this association did not remain in the adjusted logistic regression analysis.

Our study has several limitations that require consideration. Firstly, due to the retrospective nature of the study, we could not access the causative nature of the associations between subjects’ characteristics at admission and outcomes; thus, it was not possible to determine whether hypoalbuminemia and preexisting stroke preceded outcomes of interest or vice versa. Secondly, the sample size was relatively small, which reduces the external validity of the study. Thirdly, we did not perform CT chest scans for all patients who had a negative result for infiltration determined by chest X-ray but did so on the basis of clinical suspicion of pneumonia. Therefore, we suggest larger-scale, longitudinal studies be conducted to confirm our findings, to shed light on causality, and to increase understanding of the effects of pre-existing stroke and hypoalbuminemia on pneumonic COVID-19.

## 5. Conclusions

Almost a quarter of our study subjects had a pneumonic lung lesion as determined by computerized tomography, which cautions that a CT scan should be performed on COVID-19 patients negative for a pneumonic lung lesion by CXR who are suspected to be pneumonic on clinical grounds and that health care providers should place those with a potential hidden infection under medical supervision. In addition, older patients with lower albumin levels and preexisting strokes should be checked for the presence of pneumonia despite a negative CXR finding for an active lesion.

## Figures and Tables

**Table 1 healthcare-08-00518-t001:** General characteristics of the study patients.

Variable	Without PLLCT	PLLCT	*p*-Value
Number of cases	121	35	
Age, years	41.2 ± 18.4	64.1 ± 15.5	<0.001 *
Age ≥70 years	9 (7.4)	10 (28.6)	0.002 *
Sex			0.845
Male	49 (40.5)	13 (37.1)	
Female	72 (59.5)	22 (62.9)	
Resident in care facility	18 (14.9)	14 (40.0)	0.003 *
Days of hospitalization	16.42 ± 10.0	22.9 ± 12.1	0.002 *
Death	0 (0)	2 (6.2)	0.043 *
Comorbidity			
Hypertension	14 (11.6)	14 (40.0)	<0.001 *
Diabetes Mellitus	7 (5.8)	7 (20.0)	0.017 *
Dementia	3 (2.5)	5 (14.7)	0.013 *
Cardiovascular disease	0 (0)	3 (8.6)	0.011 *
Stroke	1 (0.8)	6 (17.1)	0.001 *
Malignancy	6 (5.0)	1 (2.9)	1.000
Symptoms on admission			
Asymptomatic	31 (25.6)	9 (25.7)	1.000
Cough	35 (28.9)	15 (42.9)	0.150
Sputum	19 (15.7)	6 (17.1)	0.798
Sore throat	22 (18.2)	4 (11.4)	0.445
Dyspnea	5 (4.1)	4 (11.4)	0.115
Febrile sense	30 (24.8)	12 (34.3)	0.284
Myalgia	22 (18.2)	4 (11.4)	0.445
Fatigue	6 (5.0)	1 (2.9)	1.000
Chest pain/discomfort	7 (5.8)	3 (8.6)	0.694
Nausea/vomiting	1 (0.8)	0 (0)	1.000
Diarrhea	1 (0.8)	1 (2.9)	0.400
Physical examination			
Body temperature	36.2 ±3.0	36.9 (0.6)	0.231
Systolic blood pressure	121.0 ± 14.0	130.6 ± 23.4	0.003 *
Diastolic blood pressure	78.3 ± 10.4	79.7 ± 16.4	0.550
Heart rate/min	77.6 ± 9.9	82.0 ± 20.6	0.079
breaths/min	20.1 ± 5.4	20.1 ± 1.7	0.999
O_2_ saturation	97.6 ± 1.4	96.2 ± 4.0	0.003*

Data are shown as mean ± SD or number (%). PLLCT, pneumonic lung lesion defined by computerized tomography; * denotes statistically significant set as (*p* < 0.05).

**Table 2 healthcare-08-00518-t002:** Laboratory characteristics of the study patients.

Variable	Without PLLCT	PLLCT	*p*-Value
Laboratory findings			
Leukocyte,×10^9^/L	6.4 ± 2.4	5.8 ± 2.2	0.186
Lymphocyte,×10^9^/L	1.9 ± 0.8	1.8 ± 0.7	0.405
Aspartate aminotransferase	24.0 ± 11.9	40.2 ± 58.9	0.005 *
Alanine aminotransferase	24.6 ± 17.9	32.7 ± 56.7	0.179
Creatinine	0.8 ± 0.2	0.9 ± 0.5	0.031 *
Hemoglobin	14.2 ± 1.7	13.5 ± 1.7	0.027 *
Platelet	240.2 ± 61.1	223.7 ± 100.7	0.231
Albumin, mg/dL	4.6 ± 0.4	4.1 ± 0.5	<0.001 *
Total Protein	7.3 ± 0.6	7.0 ± 0.6	0.014 *
Erythrocyte Sedimentation Rate, mm/h	11.9 ± 11.0	20.8 ± 21.1	0.002*
C-reactive protein, mg/dL	0.2 ± 0.7	1.3 ± 1.9	<0.001 *

Data are shown as mean ± SD or number (%). PLLCT, pneumonic lung lesion defined by computerized tomography; * denotes statistically significant set as (*p* < 0.05).

**Table 3 healthcare-08-00518-t003:** Univariate regression analysis of PLLCT risk factors for patients.^a.^

Variable	OR (95% CI)	*p*-Value
Age ≥ 70 years	4.978 (1.832–13.524)	0.002 *
Resident in care facility	1.455 (0.527–4.019)	0.470
Comorbidity		
Hypertension	0.530 (0.196–1.434)	0.211
Diabetes Mellitus	1.515 (0.433–5.297)	0.516
Dementia	0.649 (0.113–3.734)	0.649
Stroke	16.472 (1.372–197.725)	0.027 *
Physical examination		
Systolic blood pressure	1.022 (0.998–1.047)	0.073
O_2_ saturation	0.808 (0.598–1.091)	0.164
Laboratory findings		
Aspartate aminotransferase	1.021 (0.990–1.054)	0.190
Hemoglobin	1.175 (0.863–1.599)	0.305
Albumin	0.245 (0.070–0.862)	0.028 *
Total protein	0.974 (0.442–2.144)	0.947
Erythrocyte Sedimentation Rate	1.011 (0.983–1.040)	0.429
C–reactive protein	1.551 (1.014–2.371)	0.043 *

^a^ Adjusted for age. OR, odds ratio. CI, confidence interval. PLLCT, pneumonic lung lesion defined by computerized tomography; * denotes statistically significant set as (*p* < 0.05).

**Table 4 healthcare-08-00518-t004:** Multivariate regression analysis of PLLCT risk factors for patients.

Variable	OR (95% CI)	*p*-Value
Age ≥ 70 years	0.966 (0.251–3.720)	0.960
Stroke	11.447(1.168–112.220)	0.036 *
Albumin	0.123 (0.035–0.429)	0.001 *
C-reactive protein	1.336 (0.912–1.956)	0.137

OR, odds ratio. CI, confidence interval. PLLCT, pneumonic lung lesion defined by computerized tomography; * denotes statistically significant set as (*p* < 0.05).

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
