# Peer review of "Clinical Characteristics of the COVID-19 Patients with Pneumonia Detected by Computerized Tomography but Negative for Infiltration by X-ray"

_healthcare, 2020, doi:10.3390/healthcare8040518_

Round 1
Reviewer 1 Report
Currently, the COVID-19 pandemic has caused a worldwide increase in hospitalizations and more than 50 million people worldwide had been infected with SARS-CoV-2. In the manuscript, Acharya etc. reported the clinical characteristics of COVID-19 patients with a pneumonic lung lesion by computerized tomography (CT) that is not detected by chest X-ray (CXR). The patients with pneumonia defined by CT were older and had comorbidities such as hypertension, diabetes, cardiovascular disease, preexisting stroke, and dementia than those without. The study indicated that a CT scan should be performed on COVID-19 patients negative for a pneumonic lung lesion by CXR to explore the potential hidden infection. Generally, although the CT scan and CXR are not recommended as the only way for COVID-19 diagnosis, they might help to assess the severity of the lung infection in patients. Thus, finding out appropriate diagnostic tests is important for patients. The authors provided a clinical study that supports the use of CT scan for the diagnosis of COVID-19, which is more reliable than simple CXR. The study might significantly contribute to the diagnosis of COVID-19. However, the conclusion is largely limited by the small sample size, as only 156 COVID-19 patients were recruited in the study.
- Please add symbols to the tables to indicate the features with statistical significance.
- Table 1, p-value = 0.115* for Dyspnea. What does the “*” mean?
- The discussion is too concise. It would be better to include the current diagnostic methods and emphasize the application of imaging in COVID-19 diagnosis.
Author Response
REVIEWER 1: ROUND FIRST
Journal: Healthcare
Manuscript ID: healthcare-1015795
Title: Clinical Characteristics of the COVID-19 Patients with Pneumonia Detected by Computerized Tomography, Whose X-ray was Negative for Infiltration
Comments and Suggestions for Authors
Currently, the COVID-19 pandemic has caused a worldwide increase in hospitalizations and more than 50 million people worldwide had been infected with SARS-CoV-2. In the manuscript, Acharya etc. reported the clinical characteristics of COVID-19 patients with a pneumonic lung lesion by computerized tomography (CT) that is not detected by chest X-ray (CXR). The patients with pneumonia defined by CT were older and had comorbidities such as hypertension, diabetes, cardiovascular disease, preexisting stroke, and dementia than those without. The study indicated that a CT scan should be performed on COVID-19 patients negative for a pneumonic lung lesion by CXR to explore the potential hidden infection. Generally, although the CT scan and CXR are not recommended as the only way for COVID-19 diagnosis, they might help to assess the severity of the lung infection in patients. Thus, finding out appropriate diagnostic tests is important for patients. The authors provided a clinical study that supports the use of CT scan for the diagnosis of COVID-19, which is more reliable than simple CXR. The study might significantly contribute to the diagnosis of COVID-19. However, the conclusion is largely limited by the small sample size, as only 156 COVID-19 patients were recruited in the study.
- Please add symbols to the tables to indicate the features with statistical significance.
- Table 1, p-value = 0.115* for Dyspnea. What does the “*” mean?
- The discussion is too concise. It would be better to include the current diagnostic methods and emphasize the application of imaging in COVID-19 diagnosis.
Response: Thank you very much for your excellent comments and fruitful suggestions and we highly appreciate the reviewer’s valuable comments and suggestions. We have revised the manuscript based on reviewers’ comments and suggestions. All changes are marked with blue colored writing in this revised version of the manuscript to allow reviewers’ verifications. We have made the following point to point clarification and have addressed all comments in the manuscript.
Comment: the conclusion is largely limited by the small sample size, as only 156 COVID-19 patients were recruited in the study.
Response: Agree. The issue of sample size has been discussed as one of the limitations of the study.
Comments: Please add symbols to the tables to indicate the features with statistical significance. Table 1, p-value = 0.115* for Dyspnea. What does the “*” mean?
Response: Agree. Corrected.
Comments: The discussion is too concise. It would be better to include the current diagnostic methods and emphasize the application of imaging in COVID-19 diagnosis.
Response: Agree. Thank you very much for this comment. We have now revised it in the discussion section as suggested.
Reviewer 2 Report
This is a retrospective cross-sectional study of 156 COVID-19 patients that evaluated clinical characteristics and predictive factors of the presence of pneumonic changes on CT with negative chest X-ray. Among COVID-19 patients with normal chest X-ray, those with positive CT changes were older and had more comorbidities than those with negative CT changes. Serum albumin levels and pre-existing stroke were independent predictors of pneumonic changes on CT. The authors concluded that CT scan should be performed on COVID-19 patients with negative chest X-ray.
Major comments
- Additional information of the inclusion criteria of the study population needs to be provided. It is unclear whether this study enrolled consecutive patients who met the inclusion criteria. Given the retrospective nature of this study, I assume that the authors only included patients with CT chest who had a CT performed based on clinical suspicion and unlikely to have included all patients who had a negative chest X-ray. This is a limitation that needs to be discussed.
- Had the authors considered to include a control group of COVID patients with pneumonic changes on chest X-ray? The predictive factors of pneumonic changes on CT can be predictors of pneumonia as a group, irrespective of the diagnostic radiological modalities.
- Need to use a consistent term for describing the group of patients with pneumonic changes on CT with negative chest X-ray. The authors had used different terms, e.g. CT positive group and PLLCT, interchangeably throughout the manuscript.
- The Discussion section can be enriched by discussing alternative radiological modality for detecting pneumonic changes in patients with COVID-19, such as thoracic ultrasound, and the pros and cons of these different methods.
- Given the limitations of this study, the conclusion should be more reserved and changed from “… a CT scan should be performed…” to “… a CT chest should be considered…”.
Minor comments
- Abstract: This sentence needs to be re-phrased to convey a more accurate message, as there is not a single perfect tool for the diagnosis of pneumonia: “CXR is not usually considered a perfect tool for detecting pneumonia in COVID-19 patients”.
- Results, Line 120-121: This phrase needs to be corrected – “The mean age of those pneumonic with lung lesion..”
Author Response
REVIEWER 2: ROUND FIRST
Journal: Healthcare
Manuscript ID: healthcare-1015795
Title: Clinical Characteristics of the COVID-19 Patients with Pneumonia Detected by Computerized Tomography, Whose X-ray was Negative for Infiltration
Comments and Suggestions for Authors
This is a retrospective cross-sectional study of 156 COVID-19 patients that evaluated clinical characteristics and predictive factors of the presence of pneumonic changes on CT with negative chest X-ray. Among COVID-19 patients with normal chest X-ray, those with positive CT changes were older and had more comorbidities than those with negative CT changes. Serum albumin levels and pre-existing stroke were independent predictors of pneumonic changes on CT. The authors concluded that CT scan should be performed on COVID-19 patients with negative chest X-ray.
Response: Thank you very much for the fruitful comments and suggestions, and we greatly appreciate the reviewers' comments and suggestions. We have revised the manuscript based on reviewers’ comments and suggestions. All changes are marked with blue colored writing in this revised version of the manuscript to allow reviewers’ verifications. Below is the point to point clarification of the comments in the manuscript.
Major comments
Comment: 1. Additional information of the inclusion criteria of the study population needs to be provided. It is unclear whether this study enrolled consecutive patients who met the inclusion criteria. Given the retrospective nature of this study, I assume that the authors only included patients with CT chest who had a CT performed based on clinical suspicion and unlikely to have included all patients who had a negative chest X-ray. This is a limitation that needs to be discussed.
Response: Agree. Corrected as suggested.
Comment: 2. Had the authors considered to include a control group of COVID patients with pneumonic changes on chest X-ray? The predictive factors of pneumonic changes on CT can be predictors of pneumonia as a group, irrespective of the diagnostic radiological modalities.
Response: Agree. Unfortunately, we did not have a control group of COVID patients with pneumonic changes on chest X-ray.
Comment: 3. Need to use a consistent term for describing the group of patients with pneumonic changes on CT with negative chest X-ray. The authors had used different terms, e.g. CT positive group and PLLCT, interchangeably throughout the manuscript.
Response: Agree. Corrected.
Comment: 4. The Discussion section can be enriched by discussing alternative radiological modality for detecting pneumonic changes in patients with COVID-19, such as thoracic ultrasound, and the pros and cons of these different methods.
Response: Strongly agree. We have made modifications to the discussion as per your suggestions.
Comment: 5. Given the limitations of this study, the conclusion should be more reserved and changed from “… a CT scan should be performed…” to “… a CT chest should be considered…”.
Response: Agree. Modified.
Minor comments
Comment: 1. Abstract: This sentence needs to be re-phrased to convey a more accurate message, as there is not a single perfect tool for the diagnosis of pneumonia: “CXR is not usually considered a perfect tool for detecting pneumonia in COVID-19 patients”.
Response: Agree. Corrected.
Comment: 2. Results, Line 120-121: This phrase needs to be corrected – “The mean age of those pneumonic with lung lesion.”
Response: Agree. Corrected.
Round 2
Reviewer 2 Report
Comments have been adequately addressed.